# INSTRUCTLR: A SCALABLE APPROACH TO CREATE INSTRUCTION DATASET FOR UNDER-RESOURCED LANGUAGES

## ABSTRACT

Effective text generation and chat interfaces for low-resource languages (LRLs) remain a challenge for state-of-the-art large language models (LLMs) to support. This is mainly due to the difficulty of curating high-quality instruction datasets for LRLs, a limitation prevalent in the languages spoken across the African continent and other regions. Current approaches, such as automated translation and synthetic data generation, frequently yield outputs that lack fluency or even orthographic consistency. In this paper, we introduce InstructLR, a novel framework designed to generate high-quality instruction datasets for LRLs. Our approach integrates LLM-driven text generation with a dual-layer quality filtering mechanism: an automated filtering layer based on retrieval-augmented-generation (RAG)-based n-shot prompting, and a human-in-the-loop validation layer. Drawing inspiration from benchmarks such as MMLU in task definition, InstructLR has facilitated the creation of three multi-domain instruction benchmarks: **ZarmaInstruct-50k**, **BambaraInstruct-50k**, and **FulfuldeInstruct-50k**.

## 1 INTRODUCTION

Large language models (LLMs) are proficient in many tasks, with recent models sometimes outperforming humans, *depending on the language*. They tend to perform *substantially worse* on low-resource languages (LRLs), such as those spoken across Africa and other regions, than on higher-resource languages. This performance gap is evidently due to the limited representation of these languages in pre-training and fine-tuning datasets. Although LLMs such as GPT-4 (OpenAI et al., 2024) and Gemini (Team et al., 2024) have made progress in multilingual capabilities, many LRLs remain poorly, if at all, supported.

Existing approaches to address this gap also face major limitations. Machine translation (MT) of fine-tuning datasets from higher-resourced languages into LRLs often produces unnatural text that fails to capture language-specific nuances (Zhu et al., 2024). Synthetic data generation frequently results in hallucinated content and a lack of cultural awareness (Guo & Chen, 2024). The relatively high cost of creating human-annotated instruction data for LRLs worsens the situation.

We introduce **InstructLR**, a novel framework designed to produce high-quality instruction tuning datasets for LRLs through a combined approach that balances automation with human-in-the-loop validation. Unlike direct translation approaches that often produce unnatural outputs, InstructLR uses translation at the instruction response generation stage, where instructions—initially in a high-resource language (e.g., French)—are translated to the target LRL along with the other output components. **This allows the model to generate *contextually appropriate* responses directly in the target language (since the high resource and low resource instructions will be both embedded during the responses generation)—rather than translating complete instruction-response pairs.**

**Our contributions are as follows:**

- We propose **InstructLR**, a scalable pipeline that integrates LLM generation, RAG-based correction, and human-in-the-loop validation to produce high-quality instruction data for LRLs.

- We use this framework to create three 50k-scale, multi-domain instruction benchmarks: **ZarmaInstruct-50k**, **BambaraInstruct-50k**, and **FulfuldeInstruct-50k**—all under a CC-BY-SA 4.0 license—with links available at: **Links will be made public after the double blind review**.

- We conduct experiments comparing three training approaches: zero-shot baseline (no fine-tuning), MT-Seed baseline (fine-tuning on machine-translated instructions), and InstructLR (fine-tuning on our framework's output). This comparison aims to isolate the effectiveness of our framewok versus direct translation methods.

***Our evaluation addresses three research questions***: (RQ1) How do open-source LLMs perform on instruction-following tasks for these LRLs without fine-tuning? (RQ2) How much does fine-tuning on InstructLR datasets improve performance compared to MT baselines? (RQ3) How well do InstructLR-trained models generalize to downstream tasks?

Our study demonstrates that InstructLR enables effective instruction-following in previously unsupported languages, by achieving BLEU scores of 22.8 (Zarma), 30.1 (Bambara), and 28.9 (Fulfulde) compared to near-zero baseline performance. Furthermore, the framework reduces dataset creation costs by 88% through automated quality filtering while maintaining good linguistic quality, as validated by native speakers who preferred InstructLR outputs over machine-translation baselines in 78-84% of comparisons.

## 2 INSTRUCTLR

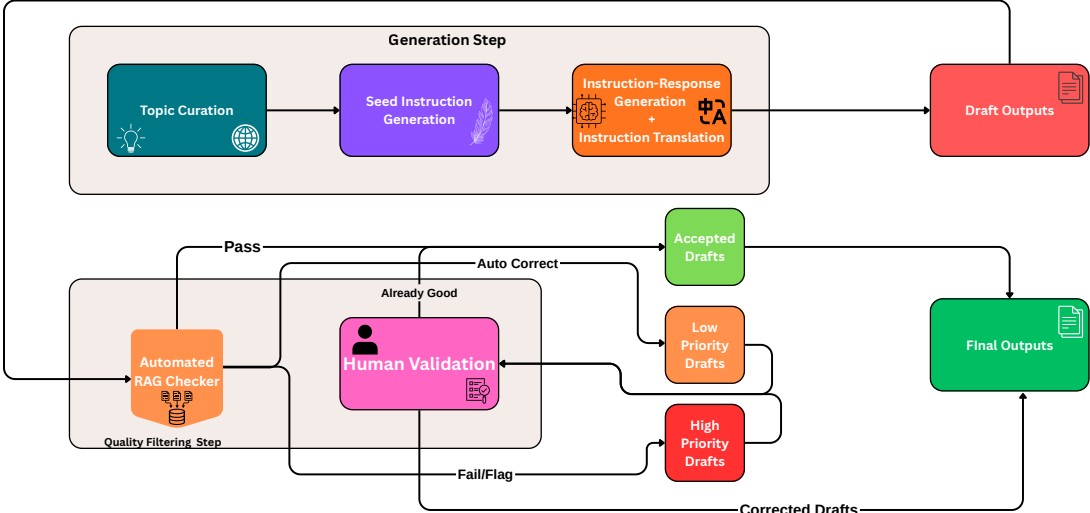

Figure 1: The InstructLR pipeline for creating high-quality instruction-tuning datasets for LRLs. The pipeline starts by the topic curation and finishes by final output.

We designed **InstructLR** (Figure 1) to assist in creating domain-specific instruction datasets for LRLs.

InstructLR consists of multiple stages—including: seed instruction, instruction-response-pair creation, automated quality checking, human validation, and the final dataset—organized as a pipeline. In this section, we describe each stage and show how they work together to produce clean instruction data.

### 2.1 SEED INSTRUCTION

**Topic Selection**    To ensure the final dataset is comprehensive and useful for training models, InstructLR starts by curating a diverse set of topics. We draw inspiration from established multi-task benchmarks like MMLU (Hendrycks et al., 2021) because they provide a structured framework of knowledge domains and reasoning skills. Our selection process targets a balanced distribution across

a wide range of areas. These include STEM fields (e.g., Physics, Mathematics, Computer Science), humanities (e.g., History, Law, Philosophy), and social sciences. The goal is to create a dataset that supports not only knowledge recall but also the development of complex reasoning abilities.

**Seed Instruction Generation**   After gathering the topic list, seed instructions are generated in a high-resource language. This approach is a necessary adaptation of the self-instruct method (Wang et al., 2023) for the LRL context. The standard self-instruct loop is technically infeasible here, as it requires a teacher model with strong generative capabilities *in the target language* to create novel instructions—a prerequisite that current models do not meet for languages like Zarma. Our method circumvents this by using the LLM for the task it can perform well (ideation in French). The choice of the high-resource language depends on its presence in the region where the target LRL is used—e.g., French-speaking countries will use French.

The seed generation process uses a modified self-instruct method, where we design an instruction generation prompt template (see Section I.1) to produce diverse, domain-appropriate instructions. We incorporate two quality control mechanisms within the prompt: (1) We add instruction diversity by using different directive verbs—e.g., explain, describe, analyze—to prevent repetitive instructions. (2) The prompt includes guidelines to avoid output that contains hallucinations, sensitive content, or falls outside the target domain.

The output is structured in a JSONL format, where each instruction is based on one topic.

### 2.2   INSTRUCTION-RESPONSE PAIRS

Once the curated set of seed instructions is prepared, the next step is generating instruction-response pairs in the target LRL. This is done using an LLM with some baseline capability—ability to generate mediocre, yet acceptable outputs—to generate content in the target LRL[1].

The LLM is prompted using a structured prompt template—(see Section I.2)—with specific guidelines to handle edge cases often encountered during translation between the higher-resource language and the target LRL, and other specifications such as the response length. **The seed instructions enable the model to translate the instructions to the LRL and generate responses directly in the LRL, informed by both the high-resource and LRL instructions—unlike MT approaches that translate pre-existing aligned segments.**.The template includes explicit constraints addressing: (1) **Word adaptation**: rules for handling technical terms, proper nouns, and domain-specific vocabulary that might not have direct equivalents in the target LRL. (2) **Prioritize understandability**: guidelines to prioritize understandability and fidelity over word-for-word translation. (3) **Language specific constraints**: language specific guidelines that cannot be generalized.

For reasoning tasks, the prompt additionally requests a chain-of-thought (CoT) component in the target LRL and ensures that the generated responses include explicit reasoning steps in the LRL.

This stage outputs **drafts** structured by key metadata fields, as shown in Table 11. Each draft includes the original instruction in the high-resource language, the translated instruction in the target LRL, the generated response in the target LRL, and, for reasoning tasks, the CoT explanation—in case of reasoning tasks—in the target LRL.

### 2.3   DUAL-LAYER QUALITY FILTERING

Raw drafts produced by an LLM often contain domain inconsistencies, fluency issues, and factual errors—particularly for LRLs with limited coverage in pretraining data. To deliver a dataset with a minimized error rate while keeping human effort affordable, we implemented a dual-layer quality pipeline that combines automated and human-driven quality assessment.

**Layer 1: Automated Quality Check**   An automated Retrieval-Augmented Generation (RAG) checker processes the drafts using a knowledge base of clean sentences, grammar rules, and glossaries of the LRL. To ground the automated quality assessment, the RAG checker retrieves relevant information to guide the LLM's correction suggestions, and ensures that every correction adheres to

---

[1]This phase only works if the chosen LLM has indeed a baseline ability to generate in the target LRL. Otherwise, the produced content would be hallucinated outputs.

lingustic rules of the LRL. With an elaborated *n*-shot prompting, it suggests corrections or flags drafts for human review. When the RAG successfully corrects a draft, it is marked as **"low priority"** for human review. If the RAG flags a draft as problematic but can not propose a correction, it is marked as **"top priority"** for human review. Drafts with no detected issues are accepted as is.

The RAG component is convenient when the LLM used for checking has moderate proficiency in the LRL. For LRLs with **"no"** LLM support, alternative strategies for the automated layer would be needed; and for LRLs where LLMs are already highly proficient, simpler prompting might suffice for the automated check.

**Layer 2: Human Validation** A team of native speakers checks drafts flagged or corrected by the RAG system. The human validation protocol varies depending on the language. However, the main objective is to assess the grammar, orthography, and fluency. All corrected and validated drafts are then formatted as JSONL.

InstructLR is designed to be language-agnostic, requiring only minimal adaptation to target a new LRL. The framework's modularity allows components to be improved or replaced depending on the context.

## 3  DATASET CREATION AND ANALYSIS

To demonstrate the effectiveness of InstructLR for generating instruction datasets, we report on our use of it to create a dataset in Zarma, a West African language spoken by over six million people (Keita et al., 2024).

### 3.1  SEED INSTRUCTION CREATION

For this stage, we selected 20 topics— listed with descriptions in Table 10—and proceeded with instruction generation. Since Zarma coexists with French in everyday usage (Keita et al., 2024), we chose French as the primary language for generating seed instructions, and a suitable model for French: the **Mistral 7b** model (Jiang et al., 2023). We then generated French instructions per topic and equally split across the topics ($\approx 5\%$ per topic).

### 3.2  DRAFT GENERATION

Once we had the curated set of French seed instructions and their associated topics, we moved on to generating the first drafts of instruction-response pairs in Zarma [2]. To achieve this, we tested several models—Gemini 2.5 Pro, GPT 4.o, and Llama 3.3 (Grattafiori et al., 2024)—to determine which one demonstrated a relatively acceptable understanding of Zarma.

We selected Gemini 2.5 Pro due to its basic understanding of Zarma. While not perfect, it outperformed other models in generating coherent Zarma texts with fewer hallucinations.

We adjusted the prompt template (see Section I.2) for Gemini and included the following specific guidelines to handle edge cases that may happen during translation between French and Zarma. These included:

**Handling of nouns and loanwords:** We instructed the model not to change proper nouns. For example, names of people, cities like Niamey, or countries like Niger should remain as they are, rendered in the target language's phonetic script. Similarly, for common French loanwords already understood in Zarma, the model was prompted to keep the existing commonly used form.

**Scientific or technical terms:** If the input text contained scientific or technical terms that do not have a direct, commonly known equivalent in Zarma—e.g., a term like "photosynthesis" or "algorithm"—the instruction was to keep the original term unchanged. The same rules apply to things like book titles, etc. The goal was to avoid the model inventing new words that would not be understood.

**Managing unknown French words:** For French words in the input that the model needed to use in the output but might not have a standard equivalent or common borrowing in the target language,

---

[2]All $50,000$ instructions were processed, and a snapshot of the outputs is shown in Table 11.

Table 1: ZarmaInstruct-50k Dataset Characteristics and Quality Assessment. [*]Percentage of top priority drafts (4,563). [†]Percentage of low priority drafts (2,535).

| (a) Dataset Characteristics | | | (b) Quality Assessment Results | | |
|---|---|---|---|---|---|
| **Metric** | **Count** | **%** | **Metric** | **Count** | **%** |
| *Instruction Distribution* | | | *Automated Filtering* | | |
| Instructions with 1–10 tokens | 1,379 | 2.76 | Total drafts processed | 50,000 | 100.00 |
| Instructions with 11–20 tokens | 27,655 | 55.31 | Accepted without correction | 42,902 | 85.80 |
| Instructions with >20 tokens | 20,966 | 41.93 | Low priority (corrected by RAG) | 2,535 | 5.07 |
| | | | Top priority (needs human review) | 4,563 | 9.13 |
| *Response Distribution* | | | *Human Validation - Top Priority* | | |
| Responses with <50 tokens | 29,833 | 59.67 | Major fluency errors | 2,574 | 56.41[*] |
| Responses with 50–100 tokens | 20,167 | 40.33 | Suffix misuse errors | 1,101 | 24.13[*] |
| Instructions with CoT reasoning | 12,500 | 25.00 | Tense consistency errors | 888 | 19.46[*] |
| *Instruction Types* | | | *Human Validation - Low Priority* | | |
| Open-ended questions | 41,957 | 83.91 | Already correct | 1,978 | 78.03[†] |
| Definition requests | 121 | 0.24 | Minor typographic adjustments | 557 | 21.97[†] |
| Explanation tasks | 5,781 | 11.56 | | | |
| List generation tasks | 2,141 | 4.28 | | | |

we allowed a process of phonetic adaptation. This means the model could **"Frenchize"** the word—writing it out in the target language's phonetic script based on its French pronunciation. A good example of this might be the French word **"politique,"** which could be written as **"politik"** in Zarma or Bambara, if that matches how such words are typically borrowed and written phonetically. This was preferred over omitting the concept or making a potentially incorrect direct translation.

### 3.3 QUALITY ASSESSMENT

**Knowledge base construction:** Our RAG checker used a knowledge base of 3,000 clean sentences from the Feriji dataset (Keita et al., 2024), 20 Zarma grammar rules each followed by examples, and bilingual glossaries, all encoded with a FAISS dense index (Douze et al., 2025). This knowledge base enabled the system to contextualize and evaluate drafts with high precision.

**Base model:** We relied on the Gemini 2.0 flash model for our RAG. Similarly to the reason of selecting Gemini 2.5 Pro for drafts generation, the choice of the model is guided by the fact that the model already has a basics understanding of the language.

The full detail of our RAG checker is explained in Section C.

After processing the 50,000-draft dataset, **4,563** drafts were flagged as top priority—a ratio of 9.126% of the dataset—while **2,535** were successfully corrected by the RAG, considered low priority (5.07%). The remaining **42,902** drafts were accepted without correction.

#### 3.3.1 HUMAN EVALUATION

**Annotator pool:** We recruited five volunteers—all native Zarma speakers with prior experience reading and writing in the language. Before starting work, annotators underwent a short training session covering: the annotation task itself, how to use the tools, and what types of corrections are acceptable. Additionally, we assessed the inter-annotator agreement using **Krippendorff's Alpha**, and obtained a score of **0.793** on 351 samples from the annotated sets. The results of the evaluation are presented in Table 1.

**Evaluation outcomes:** As shown in Table 1, among the 4,563 top-priority flagged samples, the primary issues detected were fluency problems (56.40%), followed by suffix misuse errors (24.14%) and tense consistency errors (19.46%). In the 2,535 low-priority samples, **1978** (78.028%) were already correct despite being flagged by the automated system, with the remaining **557** (21.97%) requiring only minor typographic adjustments that did not affect comprehensibility.

### 3.4 ZARMAINSTRUCT-50K DATASET

Following the InstructLR pipeline, we created ZarmaInstruct-50k, the first multi-domain instruction benchmark in the Zarma language. The dataset is composed of 50,000 instruction-response pairs covering 20 different topics (as shown in Table 10). Table 1 presents statistics of ZarmaInstruct-50k.

## 3.5 GENERALIZATION TO BAMBARA AND FULFULDE

To validate the language-agnostic nature and scalability of our framework, we applied the full **InstructLR** pipeline to two additional West African LRLs: Bambara and Fulfulde. We maintained the core methodology used for Zarma, generating **50,000** instruction-response pairs for each language using the same seed topics and French as the high-resource language. The objective was to confirm that the framework could be effectively redeployed with minimal adaptation.

The process yielded two new large-scale benchmarks: **BambaraInstruct-50k** and **FulfuldeInstruct-50k**. Initial raw drafts generated by Gemini 2.5 Pro showed error patterns comparable to those observed in Zarma—including minor fluency issues and occasional word-level hallucinations, which highlights the need for the dual-layer quality filtering mechanism to address these errors.

**More details about the generation process, raw output quality assessment, and full dataset statistics for both Bambara and Fulfulde are provided in Section** D.

## 4 EXPERIMENTS

We evaluate InstructLR through systematic experiments that assess both output quality and downstream task performance.

**Experiment Setups** We evaluate six open-source models across different parameter scales: Gemma-3-270M, Gemma-3-1B, Gemma-3-4B Team et al. (2025), Llama-3.1-8B Grattafiori et al. (2024), Mistral-7B-Instruct-v0.3, and Phi-4 Abdin et al. (2024). For each language, we split our 50k datasets into 49,000 training pairs and 1,000 held-out test pairs for evaluation.

For the baselines, We compare against two baselines; **Zero-Shot Baseline:** Each base model evaluated on test sets without fine-tuning. **MT-Seed Baseline:** To isolate the effect of our generation pipeline, we create a controlled comparison using direct MT of our French seed instructions. We fine-tune Llama-3.1-8B (our best model across all the languages experimented before the MT one)on datasets created by translating the same 50,000 French seed instructions using MADLAD-400 (Kudugunta et al., 2023)—because MADLAD is the only known model (untill this date) that supports all the three languages of this experiment. This approach avoids confusion caused by culture-specific instructions in existing datasets such as Alpaca (Taori et al., 2023).

We use unsloth (Daniel Han & team, 2023) with QLoRA (Dettmers et al., 2023) for efficient fine-tuning. Training parameters include: learning rate 2e-5, 3 epochs, with CoT responses included as supervised targets. We ensure no overlap between training and test sets.

Table 2: Results of the metric-based experiments.

| | Model | Protocol | BLEU↑ | ROUGE-L↑ | METEOR↑ |
|---|---|---|---|---|---|
| | Gemma-3-270M | Zero-Shot | 0.1±0.1 | 1.2±0.5 | 0.5±0.3 |
| | Gemma-3-270M | InstructLR | 12.5±1.8 | 18.3±2.1 | 15.1±1.9 |
| | Gemma-3-1B | Zero-Shot | 0.2±0.1 | 1.4±0.6 | 0.6±0.3 |
| | Gemma-3-1B | InstructLR | 15.8±2.0 | 22.1±2.5 | 18.4±2.2 |
| Zarma | Gemma-3-4B | Zero-Shot | 0.3±0.2 | 1.7±0.7 | 0.7±0.4 |
| | Gemma-3-4B | InstructLR | 18.2±2.2 | 25.6±2.8 | 21.3±2.5 |
| | Llama-3.1-8B | Zero-Shot | 0.3±0.2 | 1.8±0.8 | 0.8±0.4 |
| | Llama-3.1-8B | MT-Seed | 13.5±1.9 | 20.1±2.4 | 16.5±2.0 |
| | **Llama-3.1-8B** | **InstructLR** | **22.8±2.5** | **30.4±3.1** | **26.1±2.8** |
| | Mistral-7B-v0.3 | Zero-Shot | 0.2±0.1 | 1.5±0.6 | 0.6±0.3 |
| | Mistral-7B-v0.3 | InstructLR | 20.1±2.3 | 28.5±3.0 | 23.9±2.6 |
| | Phi-4 | Zero-Shot | 0.3±0.2 | 1.6±0.7 | 0.7±0.4 |
| | Phi-4 | InstructLR | 21.8±2.4 | 29.7±3.0 | 25.1±2.7 |
| | Gemma-3-270M | Zero-Shot | 0.2±0.1 | 1.1±0.5 | 0.4±0.3 |
| | Gemma-3-270M | InstructLR | 11.8±1.7 | 17.9±2.0 | 14.6±1.8 |
| | Gemma-3-1B | Zero-Shot | 0.3±0.2 | 1.6±0.7 | 0.7±0.4 |
| | Gemma-3-1B | InstructLR | 18.1±2.1 | 24.7±2.6 | 21.2±2.3 |
| Bambara | Gemma-3-4B | Zero-Shot | 0.4±0.3 | 1.9±0.8 | 0.8±0.4 |
| | Gemma-3-4B | InstructLR | 23.2±2.5 | 31.4±3.2 | 27.8±2.9 |
| | Llama-3.1-8B | Zero-Shot | 0.4±0.3 | 2.1±0.9 | 0.9±0.5 |
| | Llama-3.1-8B | MT-Seed | 21.3±2.4 | 29.8±3.0 | 25.7±2.7 |
| | **Llama-3.1-8B** | **InstructLR** | **30.1±2.9** | **39.8±3.8** | **34.5±3.4** |
| | Mistral-7B-v0.3 | Zero-Shot | 0.3±0.2 | 1.7±0.7 | 0.7±0.4 |
| | Mistral-7B-v0.3 | InstructLR | 25.8±2.7 | 34.1±3.4 | 30.2±3.1 |
| | Phi-4 | Zero-Shot | 0.4±0.3 | 1.8±0.8 | 0.8±0.4 |
| | Phi-4 | InstructLR | 27.3±2.8 | 36.5±3.6 | 32.1±3.2 |
| | Gemma-3-270M | Zero-Shot | 0.1±0.1 | 1.0±0.4 | 0.4±0.2 |
| | Gemma-3-270M | InstructLR | 10.9±1.6 | 16.8±1.9 | 13.7±1.7 |
| | Gemma-3-1B | Zero-Shot | 0.2±0.1 | 1.3±0.6 | 0.5±0.3 |
| | Gemma-3-1B | InstructLR | 16.7±2.0 | 23.1±2.5 | 19.8±2.2 |
| Fulfulde | Gemma-3-4B | Zero-Shot | 0.3±0.2 | 1.6±0.7 | 0.7±0.4 |
| | Gemma-3-4B | InstructLR | 21.8±2.4 | 29.3±3.0 | 25.9±2.7 |
| | Llama-3.1-8B | Zero-Shot | 0.2±0.2 | 1.5±0.7 | 0.6±0.4 |
| | Llama-3.1-8B | MT-Seed | 19.7±2.3 | 28.1±2.9 | 24.2±2.6 |
| | **Llama-3.1-8B** | **InstructLR** | **28.9±2.8** | **38.2±3.7** | **33.1±3.3** |
| | Mistral-7B-v0.3 | Zero-Shot | 0.2±0.1 | 1.4±0.6 | 0.6±0.3 |
| | Mistral-7B-v0.3 | InstructLR | 24.3±2.6 | 32.7±3.3 | 28.9±3.0 |
| | Phi-4 | Zero-Shot | 0.3±0.2 | 1.6±0.7 | 0.7±0.4 |
| | Phi-4 | InstructLR | 26.1±2.7 | 35.0±3.5 | 30.8±3.1 |

**Automatic Evaluation** Table 2 presents results on held-out test sets using BLEU (Papineni et al., 2002), ROUGE-L (Lin, 2004), and ME-TEOR (Banerjee & Lavie, 2005) metrics. Zero-shot performance demonstrates limitations of current

Table 4: Human quality ratings and downstream NER. The NER experiment was conducted with our best model from the automatic evaluation: **(Llama-3.1-8B with InstructLR)** (see Table 2)

(a) Human quality ratings.

| Lang | Model | Fluency ↑ | Correctness ↑ | Relevance ↑ |
|---|---|---|---|---|
| Zarma | Zero-shot | 1.2 [1.1, 1.3] | 1.1 [1.0, 1.2] | 1.3 [1.2, 1.4] |
| | MT-Seed | 2.3 [2.2, 2.5] | 2.1 [2.0, 2.3] | 2.6 [2.5, 2.7] |
| | **InstructLR** | **3.3 [3.2, 3.4]** | **2.9 [2.8, 3.1]** | **3.7 [3.6, 3.8]** |
| Bambara | Zero-shot | 1.4 [1.3, 1.5] | 1.2 [1.1, 1.3] | 1.3 [1.2, 1.4] |
| | MT-Seed | 3.0 [2.9, 3.2] | 2.7 [2.6, 2.9] | 3.3 [3.2, 3.4] |
| | **InstructLR** | **4.2 [4.0, 4.5]** | **4.0 [3.9, 4.1]** | **4.2 [4.1, 4.3]** |
| Fulfulde | Zero-shot | 1.3 [1.2, 1.4] | 1.1 [1.0, 1.2] | 1.2 [1.1, 1.3] |
| | MT-Seed | 2.8 [2.7, 3.0] | 2.5 [2.4, 2.7] | 3.1 [3.0, 3.2] |
| | **InstructLR** | **4.1 [4.0, 4.2]** | **3.8 [3.7, 4.0]** | **4.0 [3.9, 4.1]** |

(b) NER (exact match & macro-F1).

| Lang | Model | Exact Match ↑ | Macro-F1 ↑ |
|---|---|---|---|
| Zarma | Zero-shot | 9.8% [7.2, 12.7] | 21.4 [18.1, 24.7] |
| | MT-Seed | 27.6% [23.6, 31.8] | 49.3 [45.2, 53.2] |
| | **InstructLR** | **41.2% [36.8, 45.7]** | **63.8 [60.1, 67.2]** |
| Bambara | Zero-shot | 13.0% [10.1, 16.4] | 27.2 [23.9, 30.6] |
| | MT-Seed | 36.8% [32.5, 41.3] | 57.9 [54.2, 61.5] |
| | **InstructLR** | **54.4% [50.0, 58.7]** | **71.6 [68.4, 74.7]** |
| Fulfulde | Zero-shot | 12.2% [9.4, 15.6] | 25.9 [22.6, 29.3] |
| | MT-Seed | 33.0% [29.0, 37.3] | 55.2 [51.3, 58.9] |
| | **InstructLR** | **50.6% [46.2, 55.0]** | **69.1 [65.8, 72.2]** |

LLMs for these languages, with scores near zero across all models—which confirms that Zarma, Bambara, and Fulfulde are minimally or not covered by the models used for the trainings.

Fine-tuning on InstructLR datasets produces important improvements. The best-performing model (Llama-3.1-8B with InstructLR) achieves 22.8 BLEU on Zarma, 30.1 on Bambara, and 28.9 on Fulfulde. **These results demonstrate that our framework enables effective instruction-following capabilities in previously unsupported languages**.

The MT-Seed baseline underperforms InstructLR across all languages. On Zarma, InstructLR outperforms MT-Seed by 9.3 BLEU points (22.8 vs 13.5).

Table 3: Results of the human preferences experiment. The human evaluation and the MT-Seed were carried out with our best-performing model (Llama-3.1-8B with InstructLR)

| Lang | InstructLR vs. | InstructLR | Baseline | Ties |
|---|---|---|---|---|
| Zarma | Zero-Shot | 89.2% [86.1, 91.7] | 4.4% [2.9, 6.6] | 6.4% [4.5, 9.0] |
| | MT-Seed | 78.4% [74.6, 81.8] | 12.2% [9.6, 15.4] | 9.4% [7.1, 12.4] |
| Bambara | Zero-Shot | 94.0% [91.6, 95.8] | 2.4% [1.4, 4.1] | 3.6% [2.3, 5.6] |
| | MT-Seed | 83.6% [80.1, 86.6] | 8.0% [5.9, 10.7] | 8.4% [6.3, 11.1] |
| Fulfulde | Zero-Shot | 91.8% [89.0, 93.9] | 2.8% [1.6, 4.7] | 5.4% [3.6, 7.9] |
| | MT-Seed | 80.8% [77.0, 84.1] | 11.0% [8.6, 14.1] | 8.2% [6.1, 11.0] |

**Human Evaluation** We conduct comprehensive human evaluation with native speakers using our best-performing model (Llama-3.1-8B with InstructLR) across three evaluation protocols.

**-Pairwise Preference Evaluation** Two native speakers per language independently compared system outputs on 500 randomly selected prompts from our test sets. Evaluators chose between system outputs or marked ties when outputs were equivalent in quality.

Table 3 shows strong preference for InstructLR across all languages. Against zero-shot baselines, InstructLR wins in 89.2% of Zarma comparisons, 94.0% of Bambara comparisons, and 91.8% of Fulfulde comparisons. The high tie rates with zero-shot baselines (4-6%) reflect cases where both systems produced minimal or no valid output. When compared to MT-Seed baselines, InstructLR maintains substantial advantages with win rates of 78.4% (Zarma), 83.6% (Bambara), and 80.8% (Fulfulde). The lower margins against MT-Seed reflect that both systems produce fluent output, but InstructLR demonstrates higher linguistic quality and appropriateness.

**-Quality Evaluation** Native speakers rated 500 responses per protocol on three quality aspects using 5-point scales: fluency, correctness, and relevance.

Table 4 demonstrates quality advantages for InstructLR across all aspects and languages. Zero-shot baselines score poorly (1.1-1.6 range) due to their inability to generate coherent responses in these languages. MT-Seed baselines achieve moderate scores (2.1-3.3 range) but fall short of InstructLR's performance. InstructLR achieves strong scores across languages, with Bambara and Fulfulde showing particularly high ratings (4.0-4.2 range). The slightly lower Zarma scores (2.9-3.7 range) reflect the more complex grammatical structure and our evaluation criteria during the human validation process.

## 4.1 DOWNSTREAM TASK EVALUATION

To assess practical utility beyond instruction-following, we evaluate models on Named Entity Recognition (NER). We created 1,000-statement NER datasets per language with annotations for person, location, and organization entities. Models were prompted to extract entities using zero-shot prompting without task-specific fine-tuning. We evaluate using exact match accuracy and macro-averaged F1 scores.

Table 4 shows that InstructLR-trained models demonstrate strong generalization to downstream tasks. InstructLR achieves exact match scores of 41.2% (Zarma), 54.4% (Bambara), and 50.6% (Fulfulde), outperforming both zero-shot baselines (9-13% range) and MT-Seed baselines (27-37% range).

The improvements over MT-Seed baselines (13-17 percentage point gains) confirm that our quality filtering approach produces more reliable training data that enables better task generalization.

## 5 DISCUSSION

Our experimental results demonstrate that InstructLR successfully creates useful instruction datasets for under-resourced languages. The experiments confirm that models fine-tuned on our data achieve substantial improvements over both zero-shot and MT baselines. Furthermore, the performance gains across three differentlanguages—Zarma, Bambara, and Fulfulde—prove the framework's language-agnostic design.

An important component behind the framework's effectiveness is its dual-layer quality filtering mechanism. The automated RAG-based layer processes the majority of the data (85.8%) without human input, which directly enables the 88% cost reduction compared to full human annotation (see Section F). This balance makes large-scale dataset creation economically feasible. The quality of the resulting data is confirmed by the high performance on automatic metrics—where fine-tuning yields BLEU scores as high as 22.8 (Zarma), 30.1 (Bambara), and 28.9 (Fulfulde) from near-zero baselines.

Human evaluation further emphasizes these findings. Native speakers showed a strong preference for InstructLR outputs over baselines in 78-94% of comparison. Also, the model trained on ZarmaInstruct achieves a 41.2% exact match score on a zero-shot NER task, a considerable improvement over the baselines. These findings suggests the datasets from InstructLR can serve as foundational resources for real-world applications.

In sum, these findings position InstructLR as an efficient and economically friendly framework in creating multi-domain instructions dataset for LRLs, and thus opening more research opportunities for these languages.

## 6 LIMITATIONS

While InstructLR provides a robust framework for generating instruction datasets for LRLs, we acknowledge several limitations that impact its current effectiveness and scalability.

First, our framework currently relies on commercial LLMs for the initial draft generation, as these are often the only models with even a basic capability in many LRLs. This dependency introduces a cost factor that may be a challenge for researchers. Additionally, the InstructLR pipeline requires that the target LRL is at least minimally covered by an existing LLM. For languages with no current LLM support, the framework is inapplicable without significant adaptations.

Another limitation concerns the demonstrated scope of our framework. While we successfully applied it to three distinct West African languages, all three share French as a high-resource contact language. Consequently, further work is needed to validate its effectiveness for languages with different features or writing systems.

The scope of our quality assessment also presents a limitation. The automated quality assessment and human validation layers focus primarily on grammatical correctness and fluency, not on factual accuracy. Errors in the source LLM's knowledge could therefore propagate into the final datasets. Furthermore, the reliance on French seed instructions, even on general topics inspired by MMLU, could introduce a cultural bias toward Western or francophone perspectives. Finally, our human

validation relies on small annotator pools, which may not capture the full dialectal variation within the language communities.

# 7 CONCLUSION & FUTURE WORK

This paper introduces InstructLR, a framework for generating high-quality instruction datasets for low-resource languages. Our work addresses the critical data gap that limits LLM performance in these languages. Using this pipeline, we created three 50k-scale benchmarks: ZarmaInstruct-50k, BambaraInstruct-50k, and FulfuldeInstruct-50k. The framework's dual-layer quality filter, which combines RAG-based checking with human validation, effectively corrects errors while managing costs. Our experiments demonstrate that fine-tuning on these datasets enables open-source models to follow instructions in the target languages, showing significant improvements over both zero-shot and machine-translation baselines.

Future work will focus on several key areas. We aim to reduce the framework's dependency on commercial LLMs to increase its accessibility. Also, we plan to extend InstructLR to 12 new languages, including those with different high-resource contact languages and those with no existing LLM coverage. Finally, we will work to develop more sophisticated automated quality assessment techniques. These enhancements will target complex grammatical rules and aim to improve the detection of factual or cultural inconsistencies.

## 8    STATEMENT OF ETHICS

Our work aims to address an urgent gap in AI accessibility for speakers of low-resource languages. We acknowledge several ethical considerations linked to this research:

First, we recognize the importance of cultural appropriateness in generated content. While our framework incorporates human validation, we acknowledge potential limitations in capturing nuanced cultural contexts. The benchmarks reflect the expertise of our native speaker annotators but may not represent all dialectal variations or cultural perspectives within the language communities.

Second, regarding data ownership and usage rights, we emphasize that the generated instruction datasets represent content created through collaboration between automated systems and human annotators. All annotators provided informed consent for their voluntary participation, understanding how their contributions would be used in the research.

Third, we acknowledge limitations in demographic representation within our annotator pool. Our small sample of five Zarma speakers and one Bambara speaker may not represent the full diversity of these language communities. We recommend future work to expand validator diversity across age groups, regions, and educational backgrounds.

Finally, we designed our framework to minimize potential harms from generated content by incorporating multiple quality control measures. The dual-layer filtering system helps identify and remove potentially inappropriate or offensive content before inclusion in the final dataset. However, we acknowledge that no filtering system is perfect, and future users of these datasets should implement additional safeguards appropriate to their specific applications.

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

## A   USE OF LLMS

We used LLMs in several aspects of our work. First, our InstructLR pipeline, as described in Section 2, integrates LLMs for both the initial generation of seed instructions and the creation of instruction-response drafts in the target languages. In addition, we used Claude 4.1 Opus [3] to help us debugging and refining our codes for our both for training and data analysis. Finally, we used Grammarly [4] to correct grammatical errors and improve the overall readability of the manuscript.

---

[3] https://www.anthropic.com/news/claude-4
[4] grammarly.com

## B  RELATED WORK

**Instruction tuning for Low-Resource Languages**    Instruction tuning aligns LLMs with user needs by fine-tuning on task instruction data (Ma et al., 2025). Benchmarks—like FLAN, T0, etc—provide instruction datasets for LLMs to be trained on (Wei et al., 2022; Sanh et al., 2022; Wang et al., 2024; Hendrycks et al., 2021; Wang et al., 2020; 2019). However, these advances are centered on higher-resource languages—leaving LRLs with marginal coverage. This is particularly true for many African languages, due to the lack of task-specific data and the affordability of creating data. Recent work addresses this gap through multilingual instruction tuning. Muennighoff et al. (2023) showed that fine-tuning a multilingual model on English tasks can enable zero-shot instruction-following in other languages present only in pre-training data. Moreover, adding a small portion of multilingual data during fine-tuning yields further improvements on the target-language tasks (Muennighoff et al., 2023). Nevertheless, "severely" LRLs—particularly African languages—still lag behind, as the current benchmarks cover only relatively better-represented languages—such as Hausa or Swahili.

Several works provide instruction data specifically for African languages. For instance, Masakhane has produced datasets for tasks such as machine translation (MT) or named entity recognition (e.g., MasakhaNER supports 10 African languages (Adelani et al., 2021)). AfriInstruct integrates translation data (FLORES, MAFAND-MT for 16 languages), topic classification and summarization data (XL-Sum, etc), sentiment corpora (AfriSenti and NollySenti), and Masakhane benchmarks (NER, POS tagging) into a unified training set (Uemura et al., 2024). Yet, these are limited to a few African languages—not even half of the total languages present in the region. Our work addresses the need for scale-appropriate tools for building instruction datasets for LRL.

**Synthetic Instructions**    Due to the lack of human-written instruction data in most LRLs, a popular alternative is synthetic instruction generation. The self-instruct framework proposed by Wang et al. (2023) demonstrated that one can create an instruction dataset by prompting a language model with a handful of seed tasks to produce new instruction–response pairs. Following this, researchers have explored extending self-instruction to other languages. For example, Chen et al. (2024) translates the Alpaca English instructions into eight languages to compare multilingual vs. monolingual instruction tuning, and finds that even machine-translated instructions can provide cross-lingual benefits.

Also, it is important to mention the recent trend of using LLMs as annotators to reduce the cost of creating LRL data. For instance, Alhanai et al. (2024) leverage GPT-4o to automate parts of their quality assessment process by having the model score generated text on metrics such as fluency and factual consistency.

However, purely synthetic data approaches are not fully reliable in terms of quality. Model-generated instructions may contain errors, non-fluent phrasing, or cultural inappropriateness in the target LRL. Recent work highlights the need for careful control of LLM-synthesized data using strategies like rewriting the generated instructions or having multiple LLMs chat with each other to stimulate feedback dialog (Ma et al., 2025). Despite these solutions, this limitation still remains, and proves the need of human-in-the-loop approaches within these processes.

InstructLR leverages these previous approaches and combines their strengths into a unified framework for generating quality synthetic instruction data for LRLs with minimal human intervention. While self-instruction and translation approaches offer scalability, they often lack quality for LRLs. InstructLR addresses this limitation by integrating a robust LRL-aware dual-layer quality filtering process that includes RAG-based checks and human-in-the-loop validation to ensure higher fidelity and fluency.

## C  RAG-BASED CHECKER DETAILS

In this section, we provide an overview of the Retrieval-Augmented Generation (RAG) checker developed for quality assessment of Zarma text [5]. Our system combines dense retrieval with language-model analysis to detect and correct grammatical errors and to improve textual fluency.

---

[5] A mini-RAG version is available for public use at: `Linktobeprovideduponacceptance`

## C.1 System Architecture

The RAG checker integrates two primary components: a retrieval module and a generation/assessment module. The retrieval module uses a knowledge base comprising 3,000 clean Zarma sentences from the Feriji dataset (Keita et al., 2024), 20 Zarma grammar rules with examples, and bilingual glossaries. These resources were encoded with a FAISS dense index (Douze et al., 2025) for efficient semantic retrieval.

For the generation component, we used the Gemini 2.0 Flash model, selected for its understanding of Zarma linguistic structures. This model processes retrieved contextual information alongside input text to perform grammar checking and correction.

The system operates through the following workflow:

1. Input text is analyzed to identify potential error patterns.
2. Relevant grammar rules, example sentences, and vocabulary entries are retrieved from the knowledge base.
3. Retrieved context is incorporated into a prompt that guides the LLM to analyze and, if necessary, correct the text.
4. The system produces a structured assessment, including error identification and correction suggestions.

Our prompt design was important to ensure reliable performance. The prompt included instructions for recognizing proper nouns, maintaining linguistic coherence, and providing explicit reasoning for any corrections.

## C.2 Evaluation Protocols

To evaluate the RAG checker, we designed a controlled test set of 300 Zarma sentences. The test set comprised 200 sentences with injected grammatical errors, created by prompting the DeepSeek v3 (DeepSeek-AI et al., 2025) LLM to break specific Zarma grammar rules, and 100 unaltered sentences that served as a gold standard for measuring false-positive rates. Each sentence was processed through the RAG analyzer, and the system's assessments and corrections were compared with the gold references.

Table 5: **Performance metrics of the RAG-based checker on 300 Zarma test sentences**

| Metric | Value |
|---|---|
| GLEU Score | 0.8978 |
| $M^2$ Score | 0.3400 |
| False-Positive Rate | 0.0 |
| Fluency Assessment Score | 4.3/5 |

## C.3 Evaluation Results

Table 5 presents the quantitative results of the controlled test. The average GLEU score (0.8978) reflects close $n$-gram alignment with the gold corrections. The $M^2$ accuracy of 0.3400 indicates that at least one suggestion matched the gold correction exactly for 34 % of the error sentences. No false positives were recorded across the 100 correct sentences. In addition, 2 native Zarma speakers rated the outputs' fluency at 4.3/5.

## C.4 PROMPT CONFIGURATION

The checker uses the following core prompt:

---

**RAG Analyzer Prompt (evaluation configuration)**

```
You are a Zarma language expert. Analyze this potentially corrupted
Zarma sentence: ``{sentence}''
Rely primarily on your expertise in Zarma grammar and meaning.
Recognize proper nouns unless contradicted by the glossary.
Use the grammar check and glossary below as supplementary aids.
Grammar check results: {grammar_check}
Glossary information: {glossary_info}
Provide the analysis in this format:
Is the sentence correct? [Yes/No]
Reason for Incorrectness (if applicable): [Brief reason]
Corrections (if incorrect):
Option 1: [Corrected sentence with explanation]
Option 2: [Corrected sentence with explanation]
Option 3: [Corrected sentence with explanation]
```

## C.5 EXAMPLE ANALYSIS

---

**Sentence analyzed:** "Demain, a koy Niamey"
**Grammar status:** Correct (basic syntax, with caveats)

---

```
WORD BREAKDOWN:
  Demain:  Adverb, 'tomorrow' (French loanword)
  a:  3rd-person singular pronoun, 'she/he/it'
  koy:  Verb, 'to go'
  Niamey:  Proper noun, city name

LINGUISTIC INSIGHT:
  Word order:  Adheres to Zarma SVO, initial adverbs allowed.
  Tense:  Lacks future marker "ga", implying habitual / near-future action.
  Context:  Suggests "Tomorrow, she/he goes to Niamey"; "Demain" shows code-switching.

CORRECTNESS ASSESSMENT:
  Is the sentence correct?  No
  Reason:  Missing future marker for "tomorrow"; "Demain" is non-standard.

CORRECTIONS:
  Option 1:  Suba, a ga koy Niamey
  Option 2:  Suba, a koy Niamey
  Option 3:  Demain, a ga koy Niamey
```

---

```
Context sources (RAG retrieval):
  Demain:  French "demain", Zarma "suba"
  a:  French "elle", Zarma "a"
  koy:  French "aller", Zarma "koy"
```

Figure 2: Example of RAG analysis output for a single sentence.

# D    GENERALIZABILITY: ADAPTING INSTRUCTLR TO BAMBARA AND FULFULDE

To validate the adaptability and scalability of InstructLR across different languages, we applied the framework to two additional West African languages: Bambara and Fulfulde.

## EXPERIMENTAL SETUP

For these experiments, we maintained the core pipeline structure used in the Zarma implementation. We generated **50,000** instruction-response pairs for both Bambara and Fulfulde using Gemini 2.5 Pro, the same model used for Zarma, with instructions spread randomly across the 20 topics. The objective was to evaluate whether the framework could transfer to other LRLs with minimal modifications.

To assess the raw output quality and better understand the necessity of the automated filtering stage, we implemented a simplified version of the pipeline by excluding the dual-layer quality filtering mechanism. Instead, we provided a random sample of 300 draft instruction-response pairs for each language to native speakers for manual quality assessment.

## EVALUATION RESULTS

For **Bambara**, the native speaker evaluation revealed that approximately 26% of samples had minor fluency problems. These issues did not significantly impact comprehension but indicated the need for better phrasing. A more significant problem was the detection of hallucinated words in 2% of samples—one instance with a **Hindi** word and another containing a **Russian** word. Despite these issues, the remaining 72% of the samples were considered correct and understandable.

For **Fulfulde**, the evaluation showed a similar pattern, with approximately 17% of samples containing fluency errors and 1% containing hallucinated words. The errors in Fulfulde often related to its complex noun class system—something that our RAG checker could handle.

For both languages, evaluators noted that the content was easily accessible to bilingual speakers. This accessibility stems from the framework's approach to technical terminology, which remained unchanged or was adapted from French. While this ensures comprehension for bilingual speakers, monolingual speakers might face challenges with these technical concepts.

These scaled experiments with Bambara and Fulfulde demonstrate that the core instruction-response generation component of InstructLR transfers well across linguistically diverse LRLs. The presence of fluency issues and hallucinations underscores the importance of the dual-layer quality filtering approach to produce high-fidelity datasets at scale.

Table 6: **BambaraInstruct-50k Dataset Statistics.**

| Metric | Value | % or Average |
|---|---|---|
| *Instruction Characteristics* | | |
| Instructions with 1–10 **tokens** | 1,053 | 2.11% |
| Instructions with 11–20 **tokens** | 29,966 | 59.93% |
| Instructions with >20 **tokens** | 18,981 | 37.96% |
| *Response Characteristics* | | |
| Responses with <50 **tokens** | 28,346 | 56.69% |
| Responses with 50–100 **tokens** | 21,654 | 43.31% |
| Instructions with CoT reasoning | 12,500 | 25.00% |
| *Instruction Type Distribution* | | |
| Open-ended questions | 41,953 | 83.91% |
| Definition requests | 66 | 0.13% |
| Explanation tasks | 5,936 | 11.87% |
| List generation tasks | 2,045 | 4.09% |

Table 7: **FulfuldeInstruct-50k Dataset Statistics.**

| Metric | Value | % or Average |
|---|---|---|
| *Instruction Characteristics* | | |
| Instructions with 1–10 **tokens** | 4,390 | 8.78% |
| Instructions with 11–20 **tokens** | 31,273 | 62.55% |
| Instructions with >20 **tokens** | 14,337 | 28.67% |
| *Response Characteristics* | | |
| Responses with <50 **tokens** | 42,786 | 85.57% |
| Responses with 50–100 **tokens** | 7,214 | 14.43% |
| Instructions with CoT reasoning | 12,500 | 25.00% |
| *Instruction Type Distribution* | | |
| Open-ended questions | 39,765 | 79.53% |
| Definition requests | 219 | 0.44% |
| Explanation tasks | 7,431 | 14.86% |
| List generation tasks | 2,585 | 5.17% |

## E    ANNOTATOR PROTOCOL AND QUALITY ASSURANCE

The integrity of the final datasets relies partially on the quality and consistency of the human validation layer. To ensure a high standard of accuracy, we designed and implemented a structured protocol for annotator recruitment, training, and workflow management. This section provides a detailed account of that process.

### E.1    RECRUITMENT AND TRAINING

We recruited a team of native speakers for each target language. The primary validation effort for **ZarmaInstruct-50k** was conducted by a team of five annotators. For the initial quality assessments of Bambara and Fulfulde, we worked with two native speakers for each language. All participants are graduate students with a formal background in Computer Science and are fluent in both their native language and French. While none had prior formal experience in linguistic annotation, their technical background facilitated a quick adoption of the structured task requirements.

Before starting the main annotation task, all participants underwent a mandatory 40-minute training session. The session covered:

1. **Project Goals:** An overview of the project's objective to create high-quality instruction datasets and the role of human validation in correcting the nuanced errors that automated systems miss.

2. **Tooling:** A practical walkthrough of the annotation interface, which was implemented in Google Sheets for its accessibility and real-time collaboration features.

3. **Linguistic Guidelines:** A detailed review of the annotation guidelines (see Section E.3), with a focus on distinguishing between different error types.

Following the training, annotators participated in a calibration phase. During this phase, all annotators independently evaluated a common set of 50 drafts. Afterward, the team convened to discuss their decisions and resolve any disagreements.

### E.2    ANNOTATION WORKFLOW AND TOOLING

The annotation task was managed entirely within a shared Google Sheets environment. Each language had a dedicated workbook, and drafts were assigned to annotators in batches of 200. The sheet was structured with the following columns to create a clear and efficient workflow:

- `draft_id`: A unique identifier for each instruction-response pair.

- `instruction_lrl`: The original, uncorrected instruction in the target LRL, as generated by the LLM. This field was locked.

- `response_lrl`: The original, uncorrected response in the target LRL. This field was locked.

- `rag_status`: The status assigned by the automated checker (e.g., 'top_priority', 'low_priority').

- `is_correct`: A dropdown menu with two options ('Yes', 'No'). Annotators selected 'Yes' if the draft was entirely free of errors.

- `corrected_instruction`: An editable field where the annotator would provide the corrected version of the instruction, if necessary.

- `corrected_response`: An editable field for the corrected version of the response.

- `error_category`: A dropdown menu with predefined error categories (e.g., 'Fluency', 'Suffix Misuse', 'Tense Inconsistency', 'Orthography'). This structured data was essential for our error analysis.

- `comments`: An optional text field for the annotator to leave notes about ambiguous cases or complex corrections.

Annotators were instructed to first assess the draft and set the is _*correct* flag. If they selected 'No', they were then required to provide corrections in the corresponding 'corrected_' fields and select the primary error category.

### E.3 ANNOTATION GUIDELINES

To maintain consistency, all annotators adhered to a defined set of guidelines:

1. **Preserve Semantic Intent:** The primary rule was to correct linguistic errors without altering the core meaning or intent of the original French instruction. The goal was to fix the language, not the content.

2. **Prioritize Fluency and Naturalness:** Corrections should result in text that sounds natural to a native speaker. This often involved rephrasing sentences that were grammatically correct but idiomatically awkward due to literal translation.

3. **Correct All Linguistic Errors:** Annotators were tasked with identifying and fixing all grammatical, orthographic (spelling), and syntactic errors. This included issues with tense, noun-verb agreement, and the misuse of function words or suffixes.

4. **Ensure Consistent Handling of Loanwords:** Annotators followed the same rules provided to the LLM: technical terms from French were to be preserved, and other non-translatable words were to be rendered using phonetic adaptation.

### E.4 COMMON ERROR CATEGORIES AND CORRECTION EXAMPLES

During the human validation phase, several recurrent error patterns emerged. Table 8 provides illustrative examples of these common errors and the corrections applied by the annotators for the Zarma language.

### E.5 INTER-ANNOTATOR AGREEMENT (IAA)

To validate the consistency of our annotation process and the clarity of our guidelines, we measured Inter-Annotator Agreement (IAA). We calculated Krippendorff's Alpha ($\alpha$). For the Zarma dataset, a randomly selected sample of 351 drafts was annotated by all five annotators. For Bambara and Fulfulde, a smaller sample of 50 drafts was cross-annotated to validate the initial quality assessment task.

The results, presented in Table 9, show a high level of agreement for the primary Zarma annotation task and substantial agreement for the initial assessments of Bambara and Fulfulde.

The pretty high alpha score for Zarma ($\alpha$ = 0.793) indicates that the guidelines were effective and the annotators applied them. An analysis of disagreements revealed two primary sources:

Table 8: **Examples of Common Errors and Applied Corrections in Zarma.**

| Error Category | Erroneous Draft Example | Corrected Version | Rationale |
|---|---|---|---|
| **Suffix Misuse** | *Ay na hansi di.* (I saw dog.) | *Ay na **hanso** di.* (I saw **the** dog.) | The draft was missing the definite article suffix '-o'. The correction adds the suffix to make the noun 'hansi' (dog) definite, which is required by the context. |
| **Tense Inconsistency** | *Suba, a **koy** Niamey.* (Tomorrow, he/she **went** to Niamey.) | *Suba, a **ga** koy Niamey.* (Tomorrow, he/she **will go** to Niamey.) | The adverb 'Suba' (tomorrow) establishes a future context, but the verb lacks the future tense marker 'ga'. The correction inserts the marker to ensure grammatical consistency. |
| **Wrong Phrasing (Fluency)** | *Boro fo kaŋ ga ti alfa go no.* (A person who is a teacher is there.) | *Alfa fo go no.* (A teacher is there.) | The original phrasing is a literal, word-for-word translation (calque) of the French "Une personne qui est un enseignant...". The corrected version is more concise and idiomatically natural in Zarma. |
| **Orthography** | *Iri ga **barma** te.* (We will do **work**.) | *Iri ga **barna** te.* (We will do **work**.) | The word for "work" was misspelled. The correction applies the standard orthography for 'barna'. |

Table 9: **Inter-Annotator Agreement Scores**

| Language | Annotation Task | Sample Size | Krippendorff's Alpha ($\alpha$) |
|---|---|---|---|
| Zarma | Full Error Correction & Categorization | 351 | 0.793 |
| Bambara | Initial Quality Assessment (Correct/Incorrect) | 50 | 0.821 |
| Fulfulde | Initial Quality Assessment (Correct/Incorrect) | 50 | 0.637 |

- **Subjectivity in Fluency:** The most frequent source of disagreement arose from the subjective nature of fluency. One annotator might accept a phrasing as adequate, while another would suggest an alternative phrasing.
- **Dialectal Variation:** Minor disagreements occasionally rose from regional variations in vocabulary or preferred sentence structures.

In all cases of disagreement, the final version included in the dataset was determined through a majority vote. If no majority existed, a final decision was made by the lead author in consultation with the annotators.

# F    Cost Comparison

To quantify the economic efficiency of our framework, we provide a detailed cost comparison for building a **50,000-pair** LRL instruction dataset under three distinct scenarios: *LLM Only (No QC)*, *Full Human Correction*, and our proposed *InstructLR (RAG + Human)* pipeline. The analysis, summarized in Figure 3, covers both commercial API models and self-hosted open-source models, factoring in their per-token costs and estimated baseline error rates—the proportion of generated pairs requiring correction before any filtering.

Our cost model is based on the following up-to-date estimates:

- **LLM Costs:** We use an average of 75 tokens per instruction-response pair, totaling approximately 3.75 million tokens for the entire dataset. Commercial API prices are estimated at **$12/1M tokens for Gemini 2.5 Pro** and **$10/1M tokens for GPT-4o**. Self-hosted open-source models have a negligible compute cost, estimated at under $0.01/1M tokens on a single consumer GPU.

- **Human Annotation Cost:** We assume a professional annotator can review and correct a generated pair at a cost of **$0.40 per pair**. This rate was chosen based on similar study (CITATION HIDDEN FOR ANONYMITY) conducted in the past.

- **Baseline Error Rates:** Based on our initial experiments, we use the following error rates for raw generated drafts: Gemini 2.5 Pro (15%), DeepSeek-V3 (25%), GPT-4o (70%), and Llama-3-8B (95%).

The results show cost differences driven primarily by the human labor required. In a **Full Human Correction** scenario, every one of the 50,000 drafts is reviewed. This fixes the human labor cost at a substantial $**20,000** (50,000 pairs × $0.40/pair) which makes the initial LLM API cost (**$45** for Gemini) almost irrelevant to the total project budget. This high cost makes large-scale dataset creation "VERY CHALLENGING" for many research teams.

The **InstructLR pipeline** aims to address this challenge. Our dual-layer filtering process reduces the number of pairs requiring human review by approximately 88%, meaning validators only need to inspect the  6,000 pairs flagged as "top priority" or corrected by the RAG system. This slashes the human validation cost from $20,000 to just $**2,400** (6,000 pairs × $0.40/pair).

This efficiency gain has several implications. For a high-performing commercial model like Gemini 2.5 Pro, InstructLR reduces the total project cost from $20,045 (Full Correction) to $**2,445**—a saving of nearly 88%. The framework makes even models with very high error rates economically viable; a self-hosted Llama-3-8B model, despite its 95% error rate, can be used to produce a high-quality dataset for a total cost of approximately $2,400, as the automated RAG filter handles the vast majority of errors.

These results highlight that the "primary" value of InstructLR lies in its targeted reduction of human labor. By merging scalable LLM generation with an efficient, automated quality filter, our framework makes the creation of large-scale, high-quality instruction datasets for LRLs financially practical.

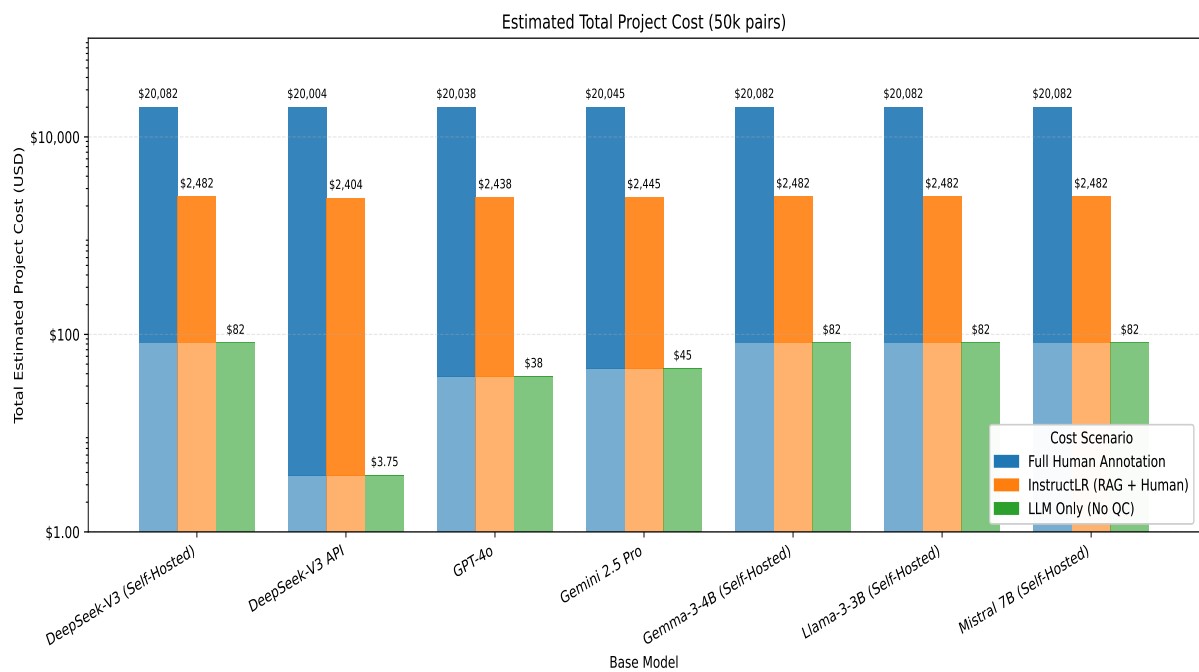

Figure 3: **Estimated total project cost** for producing 50,000 instruction–response pairs under three quality–control scenarios. Each bar shows the combined LLM compute/API cost and any required human annotation.

## G ZARMA GRAMMAR RULES

We drafted the rules below based on linguistic documentation and observations from multiple sources. The rules are not limited to these ones; however, this constitutes a baseline for future work.

RULE 1: PRONOUNS — PERSONAL PRONOUNS

Personal pronouns in Zarma are invariable across nominative, objective, and possessive cases.

- `ay` — I, me, my
- `ni` — you, your (singular)
- `a (nga)` — he, she, it; his, her, its
- `iri (ir)` — we, us, our
- `araŋ` — you (plural), your
- `i (ngey, ey)` — they, them, their

RULE 2: PRONOUNS — DEMONSTRATIVE PRONOUNS

Demonstrative pronouns indicate specific items; a `din` suffix can be added to nouns for specificity.

- `wo` — this, that
- `wey` — these, those

RULE 3: PRONOUNS — INDEFINITE PRONOUNS

Indefinite pronouns refer to non-specific entities.

- `boro` — someone, one (person)

- `hay kulu` — everything
- `hay fo` — something

RULE 4: NOUNS — DEFINITE ARTICLE

Definite articles are expressed by adding "a" or "o" to the noun based on its ending.

**Patterns:**

- Ending "a": add "a" (e.g. `zanka` → `zankaa`); exceptions: pre-1999 texts may not change.
- Ending "o": change to "a" or add "a" (e.g. `wayboro` → `waybora`).
- Ending "ko": change to "kwa" (e.g. `darbayko` → `darbaykwa`).
- Ending "e, i, u, consonant": change to "o" or add "o" (e.g. `wande` → `wando`).
- Ending "ay": change "ay" to "a" or add "o" (e.g. `farkay` → `farka` or `farkayo`).

**Examples:**

- `zanka` → `zankaa` — a child → the child
- `wayboro` → `waybora` — a woman → the woman
- `darbayko` → `darbaykwa` — a fisherman → the fisherman
- `hansi` → `hanso` — a dog → the dog
- `farkay` → `farka` — a donkey → the donkey

RULE 5: NOUNS — DEFINITE PLURAL

Definite plural is formed by replacing the definite singular vowel with "ey".

- Replace final vowel with "ey" (e.g. `zankaa` → `zankey`).
- `zankaa` → `zankey` — the child → the children
- `hanso` → `hansey` — the dog → the dogs
- `farka` → `farkey` — the donkey → the donkeys

RULE 6: NOUNS — INDEFINITE ARTICLE

No explicit indefinite article; "fo" (one) is used to specify "a certain" or "one".

- Add "fo" after noun for specificity (e.g. `musu` → `musu fo`).
- `musu` — a cat
- `musu fo` — a (certain) cat, one cat

RULE 7: NOUNS — GENDER

No grammatical gender; specific words indicate male/female for living beings.

- `alboro` — man
- `wayboro` — woman

RULE 8: VERBS — COMPLETED ACTION (PAST TENSE)

Verbs without auxiliaries indicate completed actions (past tense).

- Subject + Verb (e.g. `ay neera`).
- `ay neera` — I sold
- `a neera` — he/she sold
- `zankaa kani` — the child went to bed

RULE 9: VERBS — UNCOMPLETED ACTION (FUTURE TENSE)

Future tense uses the auxiliary "ga" before the verb.

- Subject + ga + Verb (e.g. `ay ga neera`).
- `ay ga neera` — I will sell
- `i ga neera` — they will sell

RULE 10: VERBS — CONTINUOUS ASPECT

Continuous aspect uses "go no ga" before the verb for ongoing actions.

- Subject + go no ga + Verb (e.g. `ay go no ga neera`).
- `ay go no ga neera` — I am selling
- `a go no ga neera` — he/she is selling

RULE 11: VERBS — SUBJUNCTIVE

Subjunctive uses "ma" to indicate possible actions.

- Subject + ma + Verb (e.g. `ay ma neera`).
- `ay ma neera` — I should sell
- `ni ma neera` — you should sell

RULE 12: VERBS — IMPERATIVE

Imperative uses "ma" or 'wa' before the verb, or just the verb alone.

Ma/Wa + Verb or Verb alone (e.g. `Ma haŋ` or `Haŋ`).

- `Haŋ!` — Drink!
- `Ma haŋ!` — Drink!
- `Araŋ ma di!` — You (plural) see!

RULE 13: VERBS — TO BE

The verb "to be" varies by context: "go", "ya ... no", or "ga ti".

- `A go fu` — He/she is at home
- `Ay ya alfa no` — I am a teacher
- `Nga ga ti wayboro` — She is a woman

RULE 14: VERBS — IRREGULAR VERBS

Some verbs place objects unusually (e.g. direct object before verb without "na").

- `Ay di a` — I saw him/her
- `A ne ay se` — He/she said to me

RULE 15: ADJECTIVES — QUALIFYING ADJECTIVES

Adjectives follow the noun they modify.

- `fu beeri` — a big house
- `hansi kayna` — a small dog

RULE 16: SENTENCE STRUCTURE — BASIC ORDER

Basic sentence order is Subject–Verb–Object (SVO).

- `Ay neera bari` — I sold a horse

RULE 17: SENTENCE STRUCTURE — DIRECT OBJECT

Direct object before the verb requires "na" in the past positive.

- `Ay na bari neera` — I sold a horse

RULE 18: SENTENCE STRUCTURE — INDIRECT OBJECT

Indirect object is marked with "se" after the object.

- `Ay no bari wayboro se` — I gave a horse to the woman

RULE 19: NEGATION — PAST NEGATIVE

Past negative uses "mana" after the subject.

- `Ay mana neera` — I did not sell

RULE 20: NEGATION — PRESENT/FUTURE NEGATIVE

Present/future negative uses "si" instead of "ga".

- `Ay si neera` — I do not / will not sell

# H  TOPICS SELECTED

In this section, we provide the list of topics—and a short description for each—we used for dataset creation throughout this paper.

Table 10: List of the 20 topics used for dataset generation.

| Topic | Description |
|---|---|
| General Knowledge | Includes basic factual information across diverse domains including geography, current events, etc. This category tests very basic knowledge that educated individuals are "expected" to possess. |
| Biology | Covers living organisms, their structures, functions, growth, evolution, etc. |
| Economics & Finance | Examines economic principles, financial systems, market mechanisms, etc. |
| Common Sense Reasoning | Focuses on understanding cause-and-effect relationships in familiar contexts. |
| History | Explores past events, civilizations, historical figures, their impact on contemporary society, etc. |
| Mathematics | Involves numerical computations, algebraic manipulations, geometric principles, and mathematical problem-solving. |
| Computer Science | Includes programming concepts, algorithms, data structures, software engineering, and computational thinking. It covers both theoretical computer science and practical programming applications. |
| Social Sciences & Psychology | Includes human behavior, mental processes, social interactions, and societal structures. |
| Adversarial Multi-step Reasoning | Challenges complex problem-solving abilities through multi-layered logical puzzles and sequential reasoning tasks. |
| Physics | Examines matter, energy, motion, forces, and their interactions in the physical universe. |
| Engineering | Focuses on the application of scientific and mathematical principles to design and build structures, machines, and systems. |
| Law & Ethics | Explores legal systems, ethical principles, moral reasoning, and jurisprudence. |
| Extra-difficult Reasoning | Presents highly challenging logical problems that require advanced cognitive abilities and creative problem-solving approaches. |
| Chemistry | Studies the composition, properties, and behavior of matter at the atomic and molecular level. |
| Medicine & Health | Encompasses medical knowledge, healthcare practices, disease prevention, diagnosis, and treatment approaches. |
| Business & Management | Addresses organizational management, strategic planning, leadership principles, and business operations. |
| Causal Reasoning | Tests understanding of cause-and-effect relationships, logical inference, and the ability to predict outcomes based on given conditions. |
| Sports | Covers athletic activities, rules, strategies, and sports-related knowledge including historical achievements and sporting culture. |
| Sentiment Analysis | Involves identifying and interpreting emotional tones, attitudes, and opinions expressed in text or speech. |
| Multi-sentence Comprehension | Assesses reading comprehension skills across multiple connected sentences, testing coherence understanding and information synthesis. |

# I PROMPT TEMPLATES

In this section, we show all the different prompt templates used in the InstructLR framework.

## I.1 SEED INSTRUCTIONS PROMPT TEMPLATE

---

**Seed Instruction Generation Prompt**

**Prompt**

```
Domaine : {domain}

GÉNÉREZ UNE SEULE CONSIGNE OU QUESTION EN FRANÇAIS, REPRÉSENTATIVE DE CE DOMAINE.
VOUS POUVEZ CHOISIR :
- QUESTION À CHOIX MULTIPLES (Options: A)..., B)... etc.),
- QUESTION VRAI/FAUX,
- AFFIRMATION À COMPLÉTER,
- DEMANDE DE LISTE (ex. : ``Donnez x exemples de...''),
- TÂCHE OUVERTE (CLASSIFICATION, RÉSUMÉ, EXPLICATION, EXEMPLE, ETC.),
- OU N''IMPORTE QUEL AUTRE STYLE.

CONTRAINTES :
1. RESTEZ EN 1 À 4 PHRASES.
2. NE DEMANDEZ PAS DE DESSIN, DE CHANT,
DE GÉNÉRATION D'IMAGE, NI DE RECHERCHE SUR LE WEB.
3. UTILISEZ UN VERBE UNIQUE POUR ÉVITER LA RÉPÉTITION ET MAXIMISER LA DIVERSITÉ.
4. FOURNISSEZ UNE ENTRÉE RÉALISTE (<=150 MOTS).
5. L''ENTRÉE DOIT ÊTRE SPÉCIFIQUE, SUBSTANTIELLE ET FOURNIR UN CONTENU STIMULANT.
6. NE RÉPONDEZ PAS AUX INSTRUCTIONS OU QUESTIONS
-- LIMITEZ-VOUS JUSTE À L''INSTRUCTION OU À LA QUESTION.

RENVOYEZ STRICTEMENT CE JSON :
{{
    ``instruction_fr'': ``<VOTRE INSTRUCTION>'',
    ``context_fr'': ``{domain}''
}}
'''''
```

---

## I.2 INSTRUCTION–RESPONSE PROMPT TEMPLATE

We fed the Gemini model with the prompt below to obtain an LRL instruction–response pair from a French input.

---

**LRL Instruction–Response Generation Prompt**

**System Preamble**

Vous êtes un assistant IA expert dans la génération de paires instruction–réponse pour des langues à faibles ressources, spécifiquement pour le {target_language}. Votre tâche : *(1)* générer `instr_lrl`—la version de l''instruction en {target_language}; *(2)* générer `resp_lrl`—une réponse pertinente et grammaticalement correcte en {target_language}; *(3)* pour les sujets de raisonnement (*Raisonnement de sens commun*, *Raisonnement multi-étape adversarial*, *Raisonnement extra-difficile*, *Raisonnement causal*), générer `CoT_lrl`—une explication des étapes de raisonnement en {target_language} avant la réponse, ne dépassant pas 200 mots; pour les autres sujets, `CoT_lrl` doit être "N/A". Le {target_language} est écrit en transcription phonétique.

```
\textbf{CONTRAINTES}

1. LES MOTS TECHNIQUES (SCIENCE, MÉDECINE, ETC.)
DOIVENT RESTER INCHANGÉS MAIS UTILISER LEUR
VERSION FRANÇAISE. EXEMPLE : ``ENDOMETRIOSIS'' SERA
``ENDOMÉTRIOSE''. LES TITRES DE LIVRES ET
SIMILAIRES DOIVENT RESTER INCHANGÉS.

2. SI UN MOT N'A PAS D'ÉQUIVALENT EN ZARMA,
ÉCRIVEZ SA TRANSCRIPTION PHONÉTIQUE EN FRANÇAIS.
EXEMPLE : ``POLITIQUE'' EN ZARMA SERA ``POLITIK''.

3. N'INVENTEZ PAS DE MOTS. SUIVEZ LES DIRECTIVES.

4. PAS DE TRADUCTION MOT À MOT. L'ESSENTIEL DOIT
ÊTRE FIDÈLE ET COMPRÉHENSIBLE.

5. PAS DE CRÉATIVITÉ NI D'INVENTION. RESPECTEZ
STRICTEMENT LES CONSIGNES.

6. UTILISEZ LES MOTS FRANÇAIS SI AUCUNE
TRADUCTION N'EST POSSIBLE EN ZARMA.

7. L'OBJECTIF EST UNE TRADUCTION FIDÈLE ET
COMPRÉHENSIBLE.

8. LES RÉPONSES (\verb|resp_lrl|) NE DOIVENT PAS
DÉPASSER 100 MOTS.

9. POUR LES SUJETS DE RAISONNEMENT
(\textit{Raisonnement de sens commun},
\textit{Raisonnement multi-étape adversarial},
\textit{Raisonnement extra-difficile},
\textit{Raisonnement causal}), \verb|CoT_lrl|
DOIT EXPLIQUER LES ÉTAPES DE RAISONNEMENT EN \
{target\_language\}, ÊTRE CLAIR, CONCIS, ET NE
PAS DÉPASSER 200 MOTS. POUR LES AUTRES SUJETS,
\verb|CoT_lrl| = ``N/A''.
```

**User Request**

```
{
  ``instruction_fr'': ``{user_provided_french_instruction}'',
  ``context_fr'': ``{user_provided_french_context_or_domain}'',
  ``target_language_name'': ``{target_language_full_name}'',
  ``lang'': ``{language_code}'',
  ``specific_guidelines'': [
    ``La instr_lrl DOIT être uniquement en {target_language}.'',
    ``La resp_lrl DOIT être uniquement en {target_language}.'',
    ``Pour les sujets de raisonnement, générer CoT_lrl
    expliquant les étapes de raisonnement en {target_language},
    max 200 mots. Sinon, CoT_lrl = `N/A'.,
    ``Fidélité et naturel : instr_lrl et resp_lrl doivent
    refléter l'intention.'',
    ``Conserver noms propres et emprunts établis,
    transcrits phonétiquement.'',
    ``Conserver termes techniques sans néologismes.'',
    ``Adapter phonétiquement tout mot inconnu
    (ex. 'politique' → 'politik').'',
    ``Clarté et grammaire irréprochables.''
  ]
}
```

---

```
The Output should be JSONL

{
  ``instr_fr'': ``{user_provided_french_instruction}'',
  ``instr_lrl'': ''[Generated instruction in target_language]'',
  ``resp_lrl'': ''[Generated response in target_language]'',
  ``CoT_lrl'': ''[Generated reasoning steps in target_language or 'N/A']'',
  ``topic_fr'': ``{user_provided_french_context_or_domain}'',
  ``lang'': ''{language_code}''
}
```

## J  GENERATED DATASETS SNAPSHOTS

Table 11: Snapshot of 20 instruction–response pairs from ZarmaInstruct-50k. **Snapshot of the Generated Drafts.** *Columns represent:* `instr_fr` = *French instruction,* `instr_lrl` = *Instruction in target LRL,* `resp_lrl` = *Response in LRL,* `CoT_lrl` = *Chain of Thoughts in response generation in LRL (N/A for non-reasoning topic),* `topic_fr` = *Topic name in French,* `lang` = *Language code.*

| instr_fr | instr_lrl | resp_lrl | CoT_lrl | topic_fr | lang |
|---|---|---|---|---|---|
| Quelle est la capitale du Niger ? | Niamey wo di Niger kebal? | Niamey di Niger gaba kuruso. | N/A | Connaissances générales | dje |
| Calcule 7 + 5. | 7 nda 5 baani? | 7 nda 5 ga baani 12. | N/A | Mathématiques | dje |
| Explique la loi de la gravitation. | Bay gravite kanun no? | Gravite kanun no: jano hinka deera wusu nda deera. | N/A | Physique | dje |
| Qu"est-ce qu"une molécule ? | Molekul ya mate no? | Molekul no, kaani zankakwey koyra ga. | N/A | Chimie | dje |
| Définis la photosynthése. | Fotosintez ga bay? | Fotosintez no: hanci nda saa kaani ga bi kaba. | N/A | Biologie | dje |
| Qu"est-ce qu"un algorithme ? | Algorithme ya mate no? | Algorithme no, ti hakilay hinka teeri nda namay. | N/A | Informatique | dje |
| Explique le pont suspendu. | Bay zawo suru di? | Zawo suru no, suuru sila hinka ga hin ka ti zawo fa. | N/A | Ingénierie | dje |
| Définis l"anémie. | Anemia ya mate no? | Anemia no, doro cee ka si jiri bon. | N/A | Médecine & santé | dje |
| Que signifie inflation ? | Inflation ya hin ka ti? | Inflation no, sarey gooru ka koy boori. | N/A | Économie & finance | dje |
| Définis le stress. | Stress ya mate no? | Stress no, boro hakilay ka gaama. | N/A | Sciences sociales & psychologie | dje |
| Qu"est-ce que la loi civile ? | Loi civile ya hin ka ti? | Loi civile no, koyra nda boro farilla hinka. | N/A | Droit & éthique | dje |
| Qu"est-ce qu"un business plan ? | Business plan ya mate no? | Business plan no, cilogani nda foondiray han. | N/A | Commerce & gestion | dje |
| Combien de joueurs dans un match de foot ? | Futbol kura ga boro hinka? | Futbol kura ga boro 22. | N/A | Sports | dje |
| Donne le ton du texte "A ga nafa". | Kurun "A ga nafa" seerey hin? | Kurun no seerey bara kamma, positifu. | N/A | Analyse de sentiment | dje |
| Si l"eau chauffe, que se passe-t-il ? | So dii, ko moto? | So dii, a ga buburu. | a ga buburu wa; sababaa dii. | Raisonnement causal | dje |
| Lis ces deux phrases et dis le sujet. | Ay buburu; A koy. Suje di? | Suje di "Ay" nda "A". | N/A | Compréhension multiphrases | dje |
| Pourquoi met-on un manteau en hiver ? | Kari wa, ko sabu? | Hima kura, kari ga ke boori. | Fanda kura, kari za daaba ni. | Raisonnement de sens commun | dje |
| Résous : (2 × 3) + 4. | 2 × 3 nda 4 baani? | 2 × 3 ga 6; 6 nda 4 ga 10. | mulitétape: dabari nda daaba. | Raisonnement multiétape adversarial | dje |
| Trouve le prochain nombre premier après 29. | 29 kuma, numuru kuma surey? | Numuru kuma surey ga 31. | teste divisibilité; 31 si baani. | Raisonnement extradifficile | dje |
| En quelle année le Niger fut-il indépendant ? | Niger independansi ci hinka? | Niger independansi ci 1960. | N/A | Histoire | dje |

