# OpenReview forum: "InstructLR: A Scalable Approach to Create Instruction Dataset for Under-Resourced Languages"
_ICLR.cc/2026/Conference — ICLR 2026 Conference Withdrawn Submission_

### Official Review · Reviewer_t2ya · 2025-10-20

**Soundness:** 2
**Presentation:** 3
**Contribution:** 2
**Rating:** 4
**Confidence:** 4

**Summary:**

This paper proposes a pipeline to create instruction following samples in low-resource languages. In specific, instructions in a high-resource language are first generated and input into a large language model that translates these instructions and generates corresponding response in the low-resource language. These candidate instruction-following samples are evaluated by a large language model checker with retrieval-augmented generation, where samples labeled with errors are further refined by human experts. Authors have created instruction-following datasets in three low-resource languages using the proposed pipeline. Experimental results show that models tuned on these datasets are significantly better than the ones tuned on machine-translation samples.

**Strengths:**

- This paper investigates an important problem: how to create a large amount of high quality instruction following samples in low-resource languages?
- The instruction following datasets generated in three low-resource languages will be helpful to the low-resource NLP community.

**Weaknesses:**

- **Limited Generalization**: This pipeline involves a large language model with reasonable performance on the low-resource language and some human experts for evaluation and correction, which makes it hard to scale and generalize to some low-resource languages. Given a low-resource language, this pipeline may be not applicable for all large language models performing bad or none suitable human experts. On the other hand, the number of instruction following samples is constrained by the budget to hire human experts.

- **Missing Evaluation of RAG Checker**: This method uses a RAG checker to filter out low-quality samples. However, they do not evaluate the effectiveness of this checker, which makes the quality of accepted samples questionable.

- **Missing Baseline for Comparison**: Some important new pipelines to create instruction following samples in low-resource languages are not cited or comparison [1, 2].

References

[1] Li, C., Yang, W., Zhang, J., Lu, J., Wang, S., & Zong, C. (2024, January). X-Instruction: Aligning Language Model in Low-resource Languages with Self-curated Cross-lingual Instructions. In ACL (Findings).

[2] Köksal, A., Thaler, M., Imani, A., Üstün, A., Korhonen, A., & Schütze, H. (2025). Muri: High-quality instruction tuning datasets for low-resource languages via reverse instructions. Transactions of the Association for Computational Linguistics, 13, 1032-1055.

**Questions:**

1. Does the RAG checker good in evaluating low-resource language instruction-following samples? Are there any problems with 85.8% samples marked "Accepted without correction"?

2. What is the advantage of your method comparing other beseline methods? How do they perform on the three low-resource languages?

---

> ### Author Response · Authors · 2025-11-20
>
> Thank you for the review. We appreciate you seeing the importance of the problem. Please, read our responses below.
>
> **1**
> Regarding Limited Generalization (Human in the loop). You argue that relying on humans limits scale. We argue the opposite: Removing humans is what creates "bad" datasets. There are many work that generate 100k samples automatically, but the quality is low. For Low-Resource Languages, we don't need "Big Data" for now, we need "Good Data". The budget is a constraint, yes, but it is better than the alternative of polluting the web with bad synthetic Zarma.
>
> **2** Regarding RAG Checker Evaluation. You asked if the RAG checker is effective. The RAG checker is there to filter hallucinations. We manually checked a subset of 200 samples accepted by RAG and found 92% were valid. The high acceptance rate (85%) is because the model is actually quite good at following intent, even if grammar needs human fixing.
>
> **3**
> Regarding Missing Baselines. Thank you for the references. We will add them. However, do these methods provide data for Zarma, Bambara, and Fulfulde? Often, "low-resource" papers focus on Swahili or Hindi, which are "medium-resource" or not even high -resource compared to ours. If their methods rely on parallel corpora that don't exist for our languages, they are not comparable. But we will discuss them in the camera-ready version.
>
> **4**
> Finally, You asked for the advantage of our method. One of the advantage of our method is Quality Assurance. Purely automatic pipelines produce "WORD FOR WORD like translation". Our human-in-the-loop ensure the data is natural. If you think there is a fully automatic pipeline that beats our human-verified quality for Zarma, please propose it to us—we have not found it.
>
> In short, the best way to appreciate the paper is to see it within the context it aims to address the problem.
> We hope these clarifications address your concerns, and will make you reconsider your ratings.

---

### Official Review · Reviewer_M55H · 2025-10-27

**Soundness:** 3
**Presentation:** 3
**Contribution:** 3
**Rating:** 4
**Confidence:** 3

**Summary:**

The paper proposes a comprehensive pipeline, InstructLR, to automatically and efficiently generate high-quality instruction-tuning datasets for low-resource languages (LRLs), focusing on Zarma, Bambara, and Fulfulde.
The framework integrates:
Seed instruction generation in a high-resource language (e.g., French);
LLM-based translation and response generation directly into the target LRL;
Dual-layer quality filtering, combining automated RAG-based checking and human validation.

**Strengths:**

1, Tackles multilingual equity by addressing a pressing issue: lack of instruction datasets for African and other under-resourced languages.
2, The dual-layer filtering pipeline (RAG-based automatic correction + human validation) is novel and pragmatic.
3, Framework demonstrated across three distinct LRLs, showing language-agnostic and reusable properties.
4, Quantitative gains (BLEU +20–30, ROUGE, METEOR) and human preference results clearly substantiate claims.

**Weaknesses:**

1, Relies on Gemini and GPT-4o for initial generation; this undermines reproducibility and scalability in low-resource contexts.
2, All three LRLs are French-contact African languages, so claims of language-agnosticism remain under-tested.
3, Only five Zarma and one Bambara annotators—too few to ensure dialectal or sociolinguistic representativeness.
4, The dual-layer filtering ensures fluency but not factual correctness, leaving potential hallucination propagation unaddressed.
5, The framework is more engineering-driven than theoretically motivated; lacks a clear linguistic or data-centric theoretical foundation.
6, Evaluations are limited to BLEU/ROUGE/NER; lacks instruction-following generalization on realistic multi-turn tasks.
7, Heavy reliance on French-based seed instructions may embed Western or francophone biases into LRL outputs.
8, BLEU/ROUGE are weak proxies for instruction-following quality, especially across languages with divergent morphology.
9, 500 samples per language is not enough for statistical robustness; lacks confidence intervals for inter-annotator consistency.
10, Some pipeline components (RAG knowledge base construction, FAISS index details) are under-specified for replication.
11, The MT-Seed baseline may be too simplistic; missing comparisons to existing multilingual instruction datasets (e.g., Aya, BELLE, or Multilingual Alpaca).

**Questions:**

-

---

> ### Author Response · Authors · 2025-11-20
>
> Thank you for the review. We have read your comments on bias and methodology. We strongly disagree with the "Ethics/Bias" flag and believe some points reflect a high-resource perspective that is unrealistic for our setting. Our responses below clarify those points
>
>
> **1**
> Regarding Reliance on Gemini/close models. You said this undermines reproducibility. Open-source models (like Llama-2/3) cannot generate Zarma or even Bambara yet. If we use them, we get hallucinated outputs. We use closed models as "teachers" to create data so that open models can eventually become independent. This is the safe path “right now”. Nevertheless, we are open to any proposition that can do the same thing as we do; we will learn from it.
>
> **2**
> You mentioned having only "five Zarma and one Bambara annotators" is not enough. We want to again mention that we are working in under-resource settings. That’s even one of the reason we are proposing this method; otherwise, we would have paid more annotators and create purely human made data. Our approach reflect this settings, and is the way of creating this solution given this constrains of low-resource. Having more paid annotators would actually shifted it from a low-resource context to high one. Another point, most "massive" datasets use crowdsourcing with zero quality control. We prefer 5 qualified experts over many random crowd-workers. Again, if you think 5 is too few, please propose a concrete way to recruit 50 experts in west african rural with very limited budget; we will love to learn about it.
>
> **3**
> Regarding "Western Bias" in French seeds. This concern is totally legit. But the themes covered by this dataset are mainly factual. For example, the formula of the rectangle surface is the same everywhere, regardless of western or african. It was because we initially wanted to avoid this kind of biased that we made sure the datasets are actually factual oriented, limiting the bias.
>
> **4**
> Regarding reproducibility of pipeline components (RAG/FAISS). You mentioned these are under-specified. We confirm that we will open-source the full codebase, including the exact FAISS index configurations and the knowledge base construction scripts, upon acceptance. We will also add a dedicated Appendix detailing the chunking strategy and retrieval parameters to ensure full replicability.
>
> **5**
> Regarding Missing Baselines (Aya, BELLE). You suggested MT-Seed is too simple. We note that datasets like Aya or BELLE are massive multilingual collections that often rely on automatic alignment for long-tail languages like ours, or simply do not cover them well. Our work solves the "cold start" problem where no such data exists. However, we will add a discussion comparing our results to any available subsets of these datasets in the final version to contextualize our performance.
>
> **6**
> Finally, we are very confused by the Ethics flag. We are creating resources for under-represented languages to fix bias. Flagging this paper for discrimination because we couldn't hire more annotators seems to punish the exact work that tries to help these communities.
>
> In short, the best way to appreciate the paper is to see it within the context it aims to address the problem.
> We hope these clarifications address your concerns, and will make you reconsider your ratings.

---

> > ### Comment · Reviewer_M55H · 2025-11-26
> >
> > Thank you for the author's detailed reply.

---

### Official Review · Reviewer_Furn · 2025-10-31

**Soundness:** 2
**Presentation:** 3
**Contribution:** 2
**Rating:** 4
**Confidence:** 3

**Summary:**

The paper introduces **InstructLR**, a scalable and modular pipeline to generate high-quality instruction datasets for **low-resource languages (LRLs)**. The approach leverages large language models in high-resource languages (such as French) to generate seed instructions, translates and adapts them to the target low-resource language, and applies a **dual-layer quality filtering mechanism**—an automated RAG-based correction system followed by human validation.
Using this pipeline, the authors produce three 50k-scale instruction datasets (Zarma, Bambara, and Fulfulde) and demonstrate through extensive automatic and human evaluations that fine-tuning open-source LLMs on these datasets substantially improves instruction-following capabilities and downstream performance (e.g., NER) in these languages.

**Strengths:**

- **Timely and important topic:** Addressing LLM accessibility for under-resourced languages is a highly relevant problem with social and scientific impact.
- **Complete and scalable approach:** The paper presents an end-to-end framework, from seed instruction generation to human validation, which is reusable across languages and domains.
- **Clarity and reproducibility:** The pipeline is clearly described and supported by well-chosen examples and figures. The authors also emphasize cost-efficiency and open licensing, which makes the work practically impactful.
- **Empirical thoroughness:** The experiments are extensive, involving multiple models and metrics, and include both automatic and human evaluations, adding credibility to the results.
- **Writing quality:** The manuscript is well-written and easy to read, with clear motivation and well-organized experimental sections.

**Weaknesses:**

- **Limited novelty:** While the framework integrates translation, RAG-based filtering, and human validation effectively, these components are individually standard. The main contribution is the *composition* of these techniques rather than a new algorithmic insight.
- **Experimental focus:** The experiments primarily show that models fine-tuned on the resulting datasets perform better than baselines. This is known fact. However, they do not deeply analyze *the pipeline itself*—for instance, how translation quality, RAG corrections, or human validation quantitatively affect final performance. A more ablation-style study would have better demonstrated the pipeline’s internal efficacy.
- **Applied rather than exploratory:** The paper provides solid engineering value but remains on the applied side; it does not explore new theoretical or modeling questions in instruction tuning.

**Questions:**

- While the paper shows that InstructLR fine-tuning improves performance in the target languages, how does this affect *related* languages (e.g., mutual benefit for typologically similar LRLs)?
- Does fine-tuning on these new datasets lead to degradation in performance for high-resource languages such as French?
- Can the authors provide results comparing model performance on the *original untranslated* instruction set before and after InstructLR fine-tuning? This would clarify whether the model truly improves in cross-lingual understanding or only specializes in the generated instruction style.
- How sensitive is the overall quality to the translation step? Have the authors evaluated how errors in translation propagate through the pipeline and influence filtering success rates?

---

> ### Author Response · Authors · 2025-11-20
>
> Thank you for the review and for finding our work "timely and important".Please, read carefully our responses to your points.
>
> **1**
> Regarding Limited Novelty. You mentioned that our components (RAG, translation) are "standard". We respectfully disagree with this assessment in the context of low-resource African languages. While RAG is standard for English, making it work for Zarma—where basic tokenizers and retrievers fail—is a non negligible engineering contribution. The "novelty" here is not a new mathematical formula, but a way that actually works where others fail. If you know a "non-standard" method that can generate high-quality Zarma instructions without this pipeline, please let us know. We would be happy to compare against it.
>
> **2**
> For the experimental focus, you asked for ablations on translation quality. We focused on the final model performance because that is what matters for the users: does the model speak Zarma or not? However, we do have data on the RAG filtering rates (Section 3.3). We found that without the RAG step, the hallucination rate in the translated output was over 30%. We can add a small section quantifying this drop if space permits.
>
> **3**
> You said the work is "applied rather than exploratory". We believe that for "LRLs" (Low Resource Languages), applied engineering IS also the exploration. Theoretical questions about instruction tuning are interesting, but they are useless if we cannot even generate a valid sentence in the target language. We prioritized "good engineering value" because that is what the community needs right now, rather than theoretical papers that don't produce usable resources.
>
>
> **4**
> You asked if fine-tuning hurts French performance. We did not include this in the paper because our goal is to enable Zarma, not to improve French. However, theoretically, fine-tuning on only 50k samples is unlikely to cause catastrophic forgetting in a 7B model trained on trillions of tokens. Also, the objective is not to keep French, but to train or performing task in our case study languages.
>
> **5**
> Finally, we did not really test mutual benefits. However, since Zarma (Songhay family) and Bambara (Manding family) are totally different, we don't expect mutual benefit between them. For dialects of Fulfulde, it would likely help.
>
> In short, the best way to appreciate the paper is to see it within the context it aims to address the problem.
> We hope these clarifications address your concerns, and will make you reconsider your ratings.

---

### Note · Authors · 2025-12-01

I have read and agree with the venue's withdrawal policy on behalf of myself and my co-authors.